# Incidence of critical events in the post-anesthesia care unit at a resource-limited setting in Debre Markos, Northwest Ethiopia

Abebaw Misganaw[1]*, Alaye Debas Ayenew[1], Netsanet Temesgen Ayenew[1], Enyew Fenta Mengistu[2], Baye Ashenef[2], Samrawit Nega Shiferaw[1], Getamesay Demelash Simegn[1]

1 Department of Anesthesia, School of Medicine, Debre Markos Univesity, Debre Markos, Ethiopia,
2 Department of Biomedical sciences, School of Medicine, Debre Markos Univesity, Debre Markos, Ethiopia

* abebaw_misganaw@dmu.edu.et, abebawmisganaw@gmail.com

## Abstract

### Background

Surgery and anesthesia can disrupt normal physiological function through surgical stress and residual anesthetic effects, increasing the risk of post-anesthetic complications, known as critical incidents. This study aimed to determine the incidence of critical events in the post-anesthesia care unit at Debre Markos Comprehensive Specialized Hospital, Ethiopia.

### Methods

An institution-based prospective cross-sectional study was conducted from June 1, 2024, to September 30, 2024. The sample size was determined by a single proportion formula using a prevalence of 50% and a 5% margin of error at the 95% confidence interval. The data was analyzed using SPSS version 22 for windows. Analysis was conducted using bivariable and multivariable logistic regression as needed.

### Result

Of the 422 patients, 160 (37.9%) experienced one or more critical events, with a total of 214 complications recorded. The most common critical events that occurred in the PACU were cardiovascular-related events (42%) and respiratory & airway related incidents (20%). BMI, duration of anesthesia, intraoperative complications, patient handover, PACU staff training, and ASA physical status were significantly associated with the occurrence of critical events. The odds of critical events were higher among underweight (AOR = 3.71; 95% CI: 1.27–10.79) and overweight patients (AOR = 3.05; 95% CI: 1.28–7.24). Anesthesia duration of 1–2 hours (AOR = 2.01; 95% CI: 1.06–3.81) and >2 hours (AOR = 4.11; 95% CI: 1.59–10.66) also increased

**Data availability statement:** The data is available in the supporting information file.

**Funding:** The author(s) received no specific funding for this work.

**Competing interests:** There is no competing interests among authors.

the risk. Patients with intraoperative complications had higher odds of critical events (AOR = 3.52; 95% CI: 1.88–6.58), as did those without proper handover (AOR = 3.92; 95% CI: 2.11–7.25). Increasing ASA class was associated with higher risk ASA II (AOR = 2.59; 95% CI: 1.11–6.07), ASA III (AOR = 2.86; 95% CI: 1.20–6.86), and ASA IV (AOR = 11.75; 95% CI: 2.76–50.03). Additionally, patients cared for by PACU nurses without prior PACU training were more likely to develop complications (AOR = 3.15; 95% CI: 1.73–5.72).

## Conclusion

Approximately 38% of patients experienced ≥1 critical event, mainly cardiovascular and respiratory complications. Patients who had intraoperative complications, ASA 2 to ASA 4 status, under/overweight, and those who received anesthesia for a prolonged duration were relatively at higher risk of developing critical events. There was a long time to stay in the PACU for those patients who experienced critical events.

---

## Introduction

In the post-anesthesia care unit (PACU), post-operative management is undertaken to ensure the safe recovery of patients following the completion of surgical procedures and anesthesia. This includes the identification and immediate treatment of early complications related to both anesthesia and surgery before they lead to serious adverse outcomes [1–4].

Following surgery, normal physiological functions are disrupted, resulting in the release of stress hormones such as glucagon, cytokines, cortisol, catecholamines, and antidiuretic hormone. At the same time, the residual effects of anesthetic agents and muscle relaxants may impair the body's ability to maintain homeostasis, thereby predisposing patients to post-anesthetic critical events [5].

Consequently, these critical events are major contributors to postoperative morbidity and mortality. Patients who experience such complications often require prolonged stays in the PACU or unplanned admission to higher levels of care, including the intensive care unit [5,6].

Recovery is one of the most crucial stages of anesthesia, and numerous studies have reported a high prevalence of complications involving the respiratory, cardiovascular, neurological, and airway related, surgery related (bleeding), and hypo/hyperglycemia [6,7]. The overall incidence of post-operative critical events in the PACU has varied from 0.14% to 35.2% [8–11]. This wide variation may be attributed to differences in study design, measurement methods, and definitions of critical incidents.

Notably, cardiovascular and respiratory complications account for more than 65% of all PACU complications [6,9,12]. The presence of comorbidity, type and duration of the surgery, poor communication among healthcare providers and between healthcare providers and patients, intraoperative complications, technical issues, procedure-orientated errors, misjudgments, and faults in technique are among some important contributing factors to post-operative incidents [9,11].

In developing countries, including Ethiopia, most PACU setups do not meet World Health Organization (WHO) standards, and in some hospitals, PACUs are entirely absent. As a result, patients may be transferred directly from the operating room to the ward, increasing the risk of unrecognized critical incidents. This failure to detect complications early may lead to preventable morbidity, mortality, prolonged hospital stays, and patient dissatisfaction [13].

Although several studies have examined PACU-related complications, data from Africa remain limited, where outcomes may be poorer compared to other regions [14,15]. Additionally, there is a variation in defining what constitutes post-operative critical events and the proportion of incidents among available literature worldwide. Some reports indicate that cardiovascular and respiratory complications are the leading incidents [6,9,12]. whereas others report airway-related complications as the leading events [16], In contrast, certain studies suggest that neurological complications are the most prevalent [17].

Furthermore, despite the limited information available on risk factors, there is disagreement in the literature regarding associated predictors of post-operative complications. While some studies report that American Society of Anesthesiologists (ASA) physical status classification and coexisting diseases are significant risk factors, others do not support this association [9]. Therefore, the findings of this study aim to reduce uncertainty regarding the types of post-operative critical events and their associated factors. Accordingly, this study was conducted to investigate critical events in the post-anesthesia care unit at Debre Markos Comprehensive Specialized Hospital, northwest Ethiopia.

## Method and materials

### Study area and period

The Study was conducted in Debre Markos Comprehensive Specialized Hospital from June 1, 2024 to September 30, 2024. The hospital located in the East Gojam Zone in northwest Ethiopia. The hospital has been providing services in different specialties. It has four operation theatres and on average, six cases per day have been performed. The number of patients reduced due to the study being conducted during the war in the region. Orthopedic, general, gynecologic and obstetric surgical services have been provided routinely by specialist physicians and non-physician professionals. Based on our previous observations and experiences, the hospital had no trend of recording postoperative complications in the PACU, either in hard documents or on a computer. In addition, the handover of patients between the operating room personnel and PACU nurses has been made orally without standardized checklists; indeed, the nurses record the names of patients along with the procedures done. Therefore, the oral handover of cases depends on the professionals' interest and responsibility.

### Study design and population

An institutional-based prospective cross-sectional study was conducted. The source population was all patients who received any types of anesthesia and underwent surgery in the major operating room of Debre Markos Comprehensive Specialized Hospital. The study population includes all non-ICU patients who underwent surgical procedures and who were admitted to the PACU. All patients who had elective and emergency surgery under anesthesia (both general and regional) in the major OR were included in the study. Patients who were transferred directly from the operating room to an intensive care unit were excluded from the study.

### Sampling technique and sample size determination

Sample size was determined by a single proportion formula using the prevalence of 50% because there was no previous study done in a similar setup to our study and a 5% margin of error at the 95% confidence interval using the following formula:

$$n = \frac{(z\,a/2)^2\,p\,(1-p)}{d^2}$$

Where n = sample size, z = 1.96, p =0.5, d = 0.05, CI = 95%,
And α = 5%.

$$n = \frac{(1.96)^2 \times 0.5(1 - 0.5)}{(0.05)^2} = 384$$

We added 10% of the final sample size for the non-response rate (i.e., 384 + 38 = 422). Therefore, a total sample size of 422 patients was involved in the study.

Every voluntary patient fulfilling the criteria of inclusion was selected until the required sample size was achieved by convenience sampling technique in Debre Markos Comprehensive Specialized Hospital PACU. Fig 1. Consort Flowchart.

## Study outcomes

The main outcome measure was critical events in the PACU, which was defined as any unexpected harmful events and 'near-misses', where patients did not suffer harm but which had the potential to lead to substantial negative outcome if left to progresses as a direct result of an operation. Critical events can be classified by systems; **cardiovascular related** (myocardial ischemia or infarction, premature ventricular contraction, cardiac arrest, hypotension, hypertension, tachy or bradycardia) **respiratory & airway related** (pulmonary aspiration, bronchospasm, laryngospasm, hypoxemia, re-intubation)**, CNS related** (seizure, delirium), miscellaneous (severe allergic reaction (respiratory and vascular involvement), sever transfusion reaction (respiratory and vascular involvement), bleeding…) and death [6,8,16]. Delayed PACU discharge was the secondary outcome.

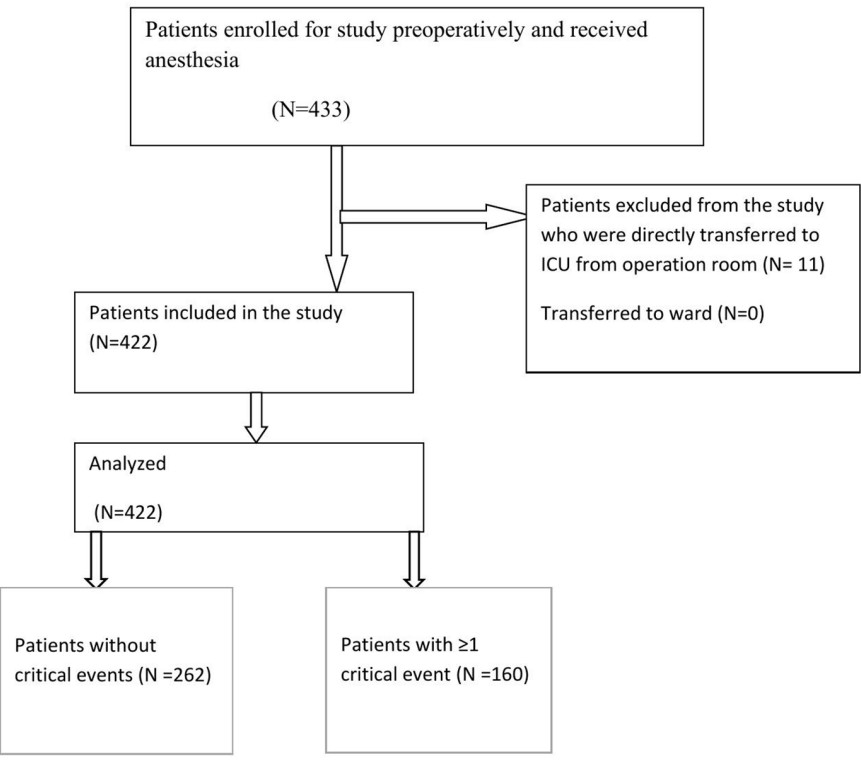

**Fig 1. Consort Flowchart.**

### Independent variables

Age, sex, American Society of Anesthesiologists physical status classification (ASA), BMI, surgical urgency, type of surgery, type of anesthesia, duration of anesthesia, patient handover practices, intraoperative complications, coexisting medical conditions, and the training status of PACU nurses (trained vs. untrained) are considered the independent variables.

### Data collection

Seven data collectors and 6 supervisor were assigned. The data collectors were BSc anesthetists and, the supervisors were six MSc anesthetists who were trained on the components of the questionnaire. The data collection procedure involved using checklists for chart data, standard tools (PAED and Adult Nursing Delirium Screening Scale for Delirium), monitoring devices used to collect vital organ-related information, as well as free texts (free narrative) to mention any critical events. The PACU was equipped with monitoring devices like electrocardiograms, heart rate monitors, arterial blood pressure monitors, and oxygen saturation devices. In the study institution, the PACU caregivers were nurses; some had some sort of training on PACU service while others hadn't. The data collectors obtained informed consent from study participants during the preoperative period. As patients arrived in the PACU, data collectors took the first data, and any critical events were documented until the patients were discharged to the ward. Since the data collectors were in the PACU only for data collection purposes, they reported any complications they found to the duty nurses to take action against complications when they weren't diagnosed timely. In the study institution, there was no standard discharging protocol, but most of the time, patients left the PACU after an average of 5 hours of stay, which was an unnecessary delay. The PACU also had many limitations in the availability of emergency equipment, trained staff, and the consultation system. Moreover, the ratio of nurses to patients is very far from a standard. The supervisors were available to control the data quality and its completeness at the end of data collection for each participant.

### Data processing and analysis

The data was entered into SPSS version 22 computer program for analysis. Descriptive statistics were used to summarize data, and tables and figures were used to display results. The association between independent factors and the outcome variable was computed using bivariable and multivariable logistic regression. Crude and adjusted odds ratio were applied to see the strength of the association for bivariable and multivariable logistic regression, respectively. Variables with a p-value less than 0.20 in the bivariable logistic regression were selected for inclusion in the multivariable logistic analysis. The Omnibus and the Hosmer & Lemeshow tests were also applied to test the model fit. Multicollinearity was checked by collinearity diagnosis and the VIF and tolerance test values were below 10 and above 0.1 respectively. A p-value of less than 0.05 was considered statistically significant.

### Data quality control and assurance

Data collectors and supervisors were trained on each item included in the study tools, objectives, relevance of the study, and rights of respondents. A pretest was conducted on 10% of the sample size at Debre Markos Comprehensive Specialized Hospital and the pretest data weren't incorporated into the main analysis. During data collection, regular supervision and follow-up were made. Investigators cross-checked for completeness, accuracy, and clarity of data on a daily basis.

### Operational definitions

**Delirium:** The presence of emergence delirium by using the Pediatric Anesthesia Emergence Delirium (PAED) Scale at ≥ 10 point in pediatrics and Nursing Delirium Screening Scale (NU-DESC) ≥2 points indicated the presence of a delirium in adults [18–21]

 **Severe hypoxemia:** O2 saturation <90% for more than 3 minutes [4,8,22]

**Hypotension:** a decrease of the systolic BP by 20% from baseline or SBP < 80mmhg [8,22,23]

**Hypertension:** an increase in systolic BP by 20–50% from baseline or SBP > 180 mmHg [8,22,24]

**Tachycardia:** Heart rate > 120 bpm in adults,160 bpm in children and greater than 180 bpm in infants considered as critical incident [22,25]

**Bradycardia:** Heart rate <50bpm for adults and < 80 for pediatrics [22].

**Stridor:** High-pitched sound during inspiration, **Wheezing:** High-pitched sound during expiration, **Hypothermia**: A temperature <36℃, **PONV**: Nausea and/vomiting during PACU stay, **Intraoperative complications**: A patient who developed any adverse events during operation [13,26,27].

**Laryngospasm:** sustained closure of the vocal cords resulting in the partial or complete loss of the patient's airway [28].

**Bronchospasm:** a rapid bronchoconstriction which leads to increased work of breathing, decreased airflow, air trapping, dynamic hyperinflation [29].

**Pain:** Moderate-to-severe pain patients having a pain score of >4 on an NRS [30].

**Underweight:** the BMI below 18.5 kg/m2 for adults and BMI < 5th percentile for children, **Healthy weight**: BMI from 18.5 to 24.9 kg/m2 for adults and BMI ≥ 5th and < 85th percentile for children. **Overweight**: BMI ≥ 25 kg/m2 for adults and BMI ≥ 85th percentiles for children. [31,32]. Obese and sever obese are included under overweight for the study purpose.

The PACU critical events were graded as per the Clavien-Dindo classification [33,34] (**Table 1**).

## Ethical consideration

Prior to the study, ethical clearance was obtained from the Research and Ethics Review Committee (RERC) of college of medicine and health science of Debre Markos University with the approval code of R/C/S/D/357/01/16 and acquiescence was also obtained from the study institution (Debre Markos Comprehensive Specialized Hospital). Moreover, full clarification about the purpose of the study was made to an authorized person of the health facility. The purpose of the study was explained to the patients who were included in the study. Written consent from the patients was obtained, and confidentiality of the information was assured by using code numbers and keeping questionnaires locked.

## Results

During the study period, 422 patients admitted to the PACU were included in the analysis. Of these, the majority were female (53.6%). The ages of patients operated on during the study period ranged from 3 to 86 years and were classified

**Table 1. The Clavien-Dindo classification.**

| Grade | Description |
|---|---|
| I | Any deviation from the normal postoperative course without the need for pharmacological treatment or surgical endoscopic and radiological interventions |
| II | Requiring pharmacological treatment with drugs other than those allowed for grade I complications. Blood transfusion and total parenteral nutrition are also included |
| III | Requiring surgical, endoscopic, or radiological intervention |
| III(a) | Intervention not under general anesthesia |
| III(b) | intervention under general anesthesia |
| IV | Life-threatening complications (including central nervous system complications) requiring intensive care unit management |
| IV(a) | Single organ dysfunction (including dialysis) |
| IV(b) | Multiorgan dysfunction |
| V | Death of a patient |

into three age groups. The proportion of complications was highest among patients older than 65 years (69.4%), indicating that the majority of patients in this group (41 out of 59) experienced critical events.

The proportion of critical events increased with higher ASA classification, with the highest proportion observed in ASA IV patients (90.6%), followed by ASA III (75.3%), ASA II (56.2%), and ASA I (19.0%). Regarding BMI, both underweight and overweight patients had a higher proportion of complications, exceeding 70.0%. (**Table 2**).

During the study period, 51.4% of patients received general anesthesia, 42.2% received regional anesthesia, and the remaining 6.4% underwent procedures under sedation. The distribution of critical events was higher among patients who received general anesthesia and sedation (38.4% and 40.7%, respectively).

As expected, the proportion of patients who experienced critical events was higher among those attended by nurses who had not received PACU training compared with those cared for by trained nurses. Differences were also observed based on anesthesia duration, with a lower proportion of complications among patients anesthetized for less than an hour. (**Table 3**).

In this study, approximately 77.7% of cases involved general surgery and gynecologic/obstetric procedures. The proportion of patients who experienced critical events across these surgical categories ranged from 35.0% to 38.4%. Among the 422 patients included, 30.8% experienced at least one intraoperative complication or a combination of complications. The proportion of patients with critical events was markedly higher among those who had intraoperative complications compared with those who did not (69.2% vs. 23.9%).

At the study institution, there was no standardized checklist or protocol for patient handover between operating room personnel and PACU nurses; instead, handover practices depended on individual professionals' initiative and were primarily oral. Oral handover between anesthetists and PACU nurses occurred in 46.2% of surgical procedures, while the remaining 53.8% of patients were transferred to the PACU without a formal oral handover. The proportion of critical events was lower among patients who received some form of handover compared with those who did not (18.9% vs. 54.1%). (**Table 4**).

Among the 422 surgical patients included in the study, 160 patients experienced a total of 214 critical events. The most common critical events observed in the PACU were cardiovascular-related complications, accounting for 42.0% of all events. Of these, 41 cases (19.15%) were hypotension, 4 cases (1.86%) hypertension, 12 cases (5.60%) shock, 5 cases (2.33%) pulmonary edema, and 27 cases (12.61%) tachycardia or bradycardia. Respiratory and airway-related complications accounted for 43 cases (20.0%) of all critical events. The most frequent respiratory event was oxygen desaturation,

**Table 2. Frequency of critical events by Socio demographic characteristic of study participants in DCSH, Ethiopia from June 1, 2024-September 30, 2024.**

| Variables | | Total number (%) | Number of patients with critical events | Incidence of critical events (% with ≥1 event, 95% CI) |
|---|---|---|---|---|
| Age in year | <15 | 53(12.6) | 18 | 33.9 (21.2, 46.6) |
| | 15-65 | 310(73.4) | 101 | 32.5 (27.3,37.7) |
| | ≥65 | 59(14) | 41 | 69.4 (57.7, 81.2) |
| Sex | M | 196(46.4) | 84 | 42.8 (35.9,49.7) |
| | F | 226(53.6) | 76 | 33.6 (27.4, 39.7) |
| ASA | 1 | 273(64.6) | 52 | 19.0 (14.3, 23.6) |
| | 2 | 48(11.4) | 27 | 56.2 (42.0,70.0) |
| | 3 | 69(16.4) | 52 | 75.3 (65.2, 85.5) |
| | 4 | 32(7.6) | 29 | 90.6 (80.5,100.7) |
| BMI | Healthy weight | 320(76) | 86 | 26.8(22.0,31.7) |
| | Underweight | 36(8.5) | 27 | 75.0 (60.8,89.1) |
| | Overweight | 66(15.5) | 47 | 71.2 (60.2,82.1) |

**Table 3. Frequency of critical events by the type of anesthesia, attended by trained nurses, and urgency of surgery in DCSH, Ethiopia from June 1, 2024-September 30, 2024.**

| Variables | | Total number (%) | Number of patients with critical events | Incidence of critical events (% with ≥1 event, 95% CI) |
|---|---|---|---|---|
| Type of anesthesia | GA | 247(51.4) | 95 | 38.4 (32.4,44.5) |
| | RA | 178(42.2) | 54 | 30.3 (23.5,37.1) |
| | Sedation | 27(6.4) | 11 | 40.7 (22.2,59.2) |
| Urgency of surgery | Elective | 289(68.5) | 110 | 38.0 (32.4,43.6) |
| | Emergency | 133(31.5) | 50 | 37.5 (29.3,45.8) |
| A patient was attended by a nurse who had PACU training | Yes | 205(48.6) | 42 | 20.4 (14.9,26.0) |
| | no | 217(51.4) | 118 | 54.3 (47.7, 61.0) |
| Duration of anesthesia per hour | <1 | 210(49.8) | 50 | 23.8 (18.0,29.5) |
| | 1-2 | 153(36.3) | 67 | 43.7 (35.9,51.6) |
| | >2 | 59(14) | 43 | 72.8 (61.5,84.2) |

**Table 4. Frequency of critical events by type of surgery, coexisting diseases, intraoperative complications, and handover of patients in PACU of DCSH, Ethiopia from June 1, 2024-September 30, 2024.**

| Variables | | Number (%) | Number of patients with critical events | Incidence of critical events (% with ≥1 event, 95% CI) |
|---|---|---|---|---|
| Type of surgery | General surgery | 195(46.2) | 75 | 38.4 (31.6,45.2) |
| | Gyni/obstetric surgery | 133(31.5) | 51 | 38.3 (30.0,46.6) |
| | Orthopedic surgery | 54(12.8) | 20 | 37.0 (24.1,49.9) |
| | Pediatric surgery | 40(9.5) | 14 | 35.0 (20.2,49.7) |
| Intraoperative complications | No | 292(69.2) | 70 | 23.9 (19.0,28.8) |
| | Yes | 130(30.8) | 90 | 69.2 (61.2,77.1) |
| Handover of patients | Oral handover | 195(46.2) | 37 | 18.9 (13.4, 24.4) |
| | No handover | 227(53.8) | 123 | 54.1 (47.6,60.5) |

followed by apnea, laryngospasm, bronchospasm, and stridor. Inadequately treated or untreated pain occurred in 34 cases (15.88%). Bleeding and anemia were reported in 20 cases (9.34%). The remaining 28 critical events (13.0%) included postoperative nausea and/or vomiting, hypothermia or shivering, delirium or convulsions, and hypo or hyperglycemia. More than half of the critical events (52.9%) occurred within the first hour of arrival in the PACU, and approximately 80.0% occurred within the first two hours. During the study period, two deaths were recorded in the PACU. Fig 2. Critical events occurred in the PACU in DCSH, Ethiopia from June 1, 2024- September 30, 2024.

## Factors associated with the incidence of PACU critical events

The overall incidence of critical events in the PACU was 37.9%. Cardiovascular complications were the most frequent (41.6%), followed by respiratory complications (20.1%). Both bivariable and multivariable logistic regression analyses were conducted to assess associations between independent variables and the outcome. Variables with a p-value < 0.20 in the bivariable analysis were included in the multivariable model, while those with p ≥ 0.20 were excluded.

The variables that met the bivariable cutoff included age, BMI, ASA physical status, type of anesthesia, intraoperative complications, duration of anesthesia, patient handover, and PACU nurse training status. The Omnibus test of model coefficients indicated that the model was statistically significant ($\chi^2 = 236.37$, df = 14, N = 422, p < 0.001), demonstrating that

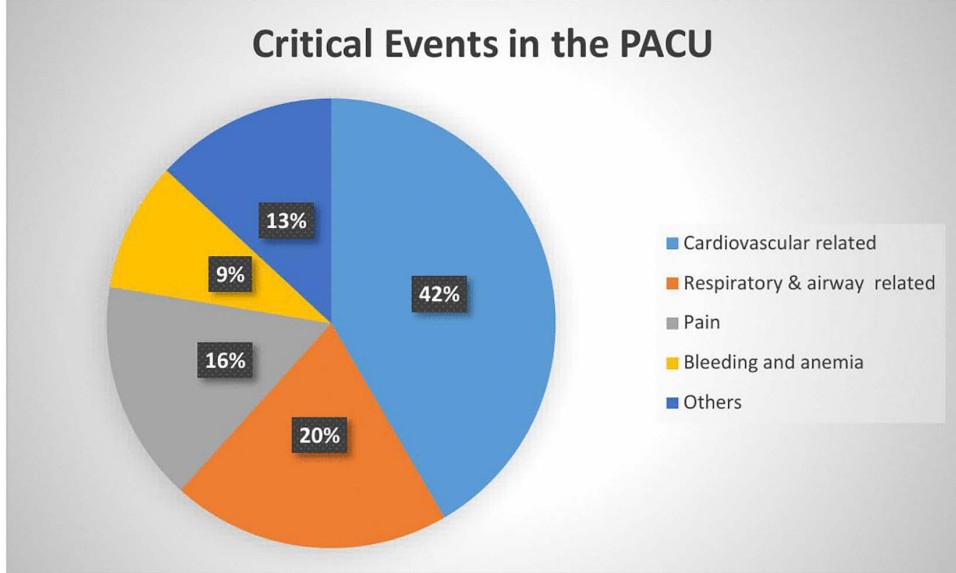

**Fig 2. Critical events occurred in the PACU in DCSH, Ethiopia from June 1, 2024–September 30, 2024.**

the predictors significantly improved model fit compared with the null model. The Hosmer–Lemeshow goodness-of-fit test showed no evidence of poor fit ($\chi^2 = 13.88$, df = 8, p = 0.085), indicating adequate model fit.

Multivariable logistic regression analysis revealed that the odds of developing critical events were significantly higher among underweight patients (AOR = 3.71; 95% CI: 1.27–10.79; p = 0.005) and overweight patients (AOR = 3.05; 95% CI: 1.28–7.24; p = 0.005). Anesthesia duration of 1–2 hours (AOR = 2.01; 95% CI: 1.06–3.81; p = 0.007) and more than 2 hours (AOR = 4.11; 95% CI: 1.59–10.66; p = 0.007) were also associated with increased risk. Patients who experienced intraoperative complications had significantly higher odds of critical events (AOR = 3.52; 95% CI: 1.88–6.58; p < 0.001), as did patients transferred to the PACU without proper handover (AOR = 3.92; 95% CI: 2.11–7.25; p < 0.001). Increasing ASA physical status was associated with progressively higher risk: ASA II (AOR = 2.59; 95% CI: 1.11–6.07; p = 0.001), ASA III (AOR = 2.86; 95% CI: 1.20–6.86; p = 0.001), and ASA IV (AOR = 11.75; 95% CI: 2.76–50.03; p = 0.001). Additionally, patients cared for by PACU nurses without prior PACU training were more likely to experience critical events (AOR = 3.15; 95% CI: 1.73–5.72; p < 0.001) (Table 5).

### Effect of critical events on PACU discharge time

As a secondary outcome, the association between critical events and delayed PACU discharge was assessed. During the study period, 93 patients did not meet discharge criteria after two hours in the PACU, of whom 88.2% had experienced critical events. There was a statistically significant association between the occurrence of critical events and delayed PACU discharge (AOR = 23.98; 95% CI: 12.17–47.28; p < 0.001).

### Discussion

Although reviewing critical events and setting preventive strategies during the perioperative period is one way of ensuring patient safety, there are limitations to international or national protocols, guidelines, or other forms of critical incident reporting and handling systems. Specially, in recovery rooms with poor infrastructure (inadequate manpower, untrained caregivers, lack of equipment, and shortages of material resources), it is difficult to review critical incidents as part of

**Table 5. Association between incidence of critical events and different variables in the PACU in DCSH, Ethiopia from June 1, 2024- September 30, 2024.**

| Predictors of critical events | | Incidence of critical events | COR(95%CI) | P value | AOR(95% CI) | P value |
|---|---|---|---|---|---|---|
| Age(year) | <15 | 33.9 | 1.00 | <0.001 | 1.00 | 0.31 |
| | 15 −65 | 32.5 | 0.94(0.50,1.74) | | 0.71(0.29,1.73) | |
| | ≥ 65 | 69.4 | 4.4(2.70,9.20) | | 1.38(0.44,4.45) | |
| ASA | 1 | 19.0 | 1.00 | <0.001 | 1.00 | 0.001 |
| | 2 | 56.2 | 5.46(2.86,10.41) | | 2.59(1.11, 6.07) | |
| | 3 | 75.3 | 13.00(6.95,24.29) | | 2.86(1.20,6.86) | |
| | 4 | 90.6 | 41.08(12.05,140.05) | | 11.75(2.76,50.03) | |
| BMI | Healthy weight | 26.8 | 1.00 | <0.001 | 1.00 | 0.005 |
| | Underweight | 75.0 | 8.16(3.69,18.05) | | 3.71(1.27,10.79) | |
| | Overweight | 71.2 | 6.73(3.74,12.11) | | 3.05(1.28, 7.24) | |
| Type of anesthesia | RA | 30.3 | 1.00 | 0.006 | 1.00 | 0.19 |
| | GA | 38.4 | 1.78(1.17, 2.71) | | 1.34(0.70, 2.53) | |
| | Sedation | 40.7 | 1.57(0.68,3.62) | | 2.87(0.89, 9.26) | |
| Duration of anesthesia | ≤ 1hrs | 23.8 | 1.00 | <0.001 | 1.00 | 0.007 |
| | 1-2hrs | 43.7 | 2.49(1.58, 3.91) | | 2.01(1.06,3.81) | |
| | >2hrs | 72.8 | 8.60(4.46, 16.57) | | 4.11(1.59,10.66) | |
| Intraoperative complications | No | 23.9 | 1.00 | <0.001 | 1.00 | <0.001 |
| | Yes | 69.2 | 7.13(4.50, 11.29) | | 3.52(1.88, 6.58) | |
| Handover of patients in the PACU | Oral handover | 18.9 | 1.00 | <0.001 | 1.00 | <0.001 |
| | No handover | 54.1 | 5.05(3.24, 7.86) | | 3.92(2.11,7.25) | |
| Was a patient attended by a nurse who had PACU training | Yes | 20.4 | 1.00 | <0.001 | 1.00 | <0.001 |
| | No | 54.3 | 4.62(3.00,7.12) | | 3.15(1.73,5.72) | |

1.00 Reference/indicator.

routine care, even though there is a high probability of a high incidence of such events. The rate of critical incident reports in previous studies varies widely; this is due to many reasons, such as differences in the definitions of complications, variations in healthcare systems among institutions, failure to recognize complications, study methodologies, and other contributing factors.

This study aimed to assess the incidence of critical events and to identify predictor factors associated with their occurrence. Although the incidence of postoperative critical events has been investigated in both developed and developing countries, there is substantial variability in reported incidences and predictor factors among studies. In this study, approximately 38% of patients developed postoperative critical events. The most frequently occurring events were cardiovascular-related, comprising 42% of all critical events, followed by respiratory-related complications (20%). Specifically, hypotension, tachyarrhythmia, stridor, anemia, shock, and desaturation occurred frequently. During the study period, the majority (71.5%) of critical events that occurred in the PACU were preventable. Many of these events could have led to patient deaths if they had not been detected and managed in a timely manner.

Hajnour M.S. et al. reported that the overall incidence of PACU critical events was 9.7%, with cardiovascular-related complications being the most frequently occurring adverse events (56%), followed by hypothermia (54%) [11]. Compared to many other studies, the findings of this survey appeared favorable. However, the possible reason for the low incidence

of complications may be the timing of measurement; critical incidents were assessed at the end of the PACU stay, whereas in our study, most critical events (80%) occurred within the first two hours of recovery.

Alghanem S.M. et al. reported that cardiovascular and respiratory-related critical events were the major causes of mortality, accounting for 33.75% and 30%, respectively, although the overall rate of complications was very low (0.14%) [35]. The extremely low rate compared to other reports may be explained by their classification of complications into minor and major, with minor complications excluded from their analysis. Chen M. et al. showed that hypertension was the most frequently occurring complications, accounting for 21.5% of PACU complications, followed by hypoxemia (7.25%) and airway obstruction (6.87%) [22]. A Multicenter study on perioperative anesthesia related adverse events in Thailand, reported that the most frequently occurred critical incident was cardiac arrhythmia requiring treatment, accounting for approximately 25% of all complications. This was followed by desaturation (24%), death within 24 hours (20%), cardiac arrest (14%), and re-intubation (10%) [4]. Unlike our study, this study reported more severe critical events, such as cardiac arrest and death, with relatively higher incidence. This difference may be attributed to the inclusion of high-risk surgical procedures, such as cardiac and thoracic operations, as well as complications occurring during the intraoperative period. In agreement with our study, Bruins S.D. et al. reported that the majority of critical incidents occurring in the PACU were cardiovascular and respiratory-related, comprising 63.9% of all complications. The urgency of surgery (emergency versus elective) was not associated with a higher incidence of complications. However, unlike our findings, they reported that patients who underwent general surgery were more likely to develop critical events compared to those undergoing other procedures [6]. The reason for this discrepancy is not clear from our data but may relate to differences in case mix or definitions.

Puchissa O. et al. reported that the most common (81%) critical incidents at the recovery unit were respiratory-related events, which were oxygen desaturation and re-intubation [36]. In that study, cardiovascular complications like hypotension, hypertension, shock, tachyarrhythmia and bradyarrhythmia were ignored, which might account for the lower incidence of cardiovascular complications. Unlike our study, Ali et al. reported that respiratory-related critical incidents were the maximum (20.7%), followed by cardiovascular complications (12.3%), with most critical incidents occurring within the first hour of the PACU stay [12]. Although not explicitly stated, differences in the definitions of critical events may account for the variation between their findings and ours.

Similar to our findings, many studies also reported that cardiovascular and respiratory critical events are common complications [4,6,11,22,35,12] but Nourizadeh M. et al. mentioned that neurologic-related complications are the most common accounting for 47.7% of all complications, while cardiovascular and respiratory-related events accounted for 34.7% and 10.8%, respectively [17]. The reason neurologic complications such as confusion and agitation were most prevalent is not well understood but may be related to anesthetic agents, premedication, or other unreported factors. In another study by Liu S. et al. reported that the most frequent critical incidents in the PACU were pain (26.9%), followed by cardiovascular-related (15.1%) and respiratory-related (14.8%) incidents [13].

In our study, the presence of intraoperative complications, the duration of anesthesia, BMI, ASA status, handover of patient, and the status of training of nurses on PACU were significantly associated with the incidence of critical events. Dharap S.B. et.al. reported that higher ASA grade, higher BMI, prolonged surgical duration, and intraoperative complications were significant associated factors, consistent with our findings [33]. Age, sex, history of hypertension, and duration of anesthesia were independent risk factors for postoperative hypertension [22]. Similar to our study, prolonged operative time was associated with the PACU complications [15]. Charuluxananan S. et al. identified factors associated with critical incidents, including ineffective communication, inadequate supervision, lack of vigilance, inexperience, and inappropriate decision-making [4]. However, the methods used to measure these factors were not clearly described.

We confirmed that the occurrence of critical events in the PACU delayed patient discharge. Similarly, Huda et al. reported that critical incidents, mainly cardiovascular and respiratory-related, prolonged the length of PACU stay [37]. Bruins S.D. et al. also found that critical incidents prolonged PACU stay and have impacts on healthcare utilization [6]. Liu S.

 

et al. reported that patients with critical incidents had a longer PACU stay (61.50 ± 44.40 minutes) compared with patients without critical incidents (28.50 ± 19.40 minutes) [13].

Based on the perioperative review of major complications and the definition proposed by Alghanem S.M. et al. [35], 43% of critical events in our study were major or sever critical complications. These events led, or could have led, to major disability or death unless immediate management was provided. According to the Clavien–Dindo classification of critical incidents, 7.9% of critical events in our study were classified as Grade IV, 45% as Grade II and III, and the remaining 47.1% as Grade I [34]. Similar to our findings, Dharap S.B. et al. reported that 31.5% of patients experienced postoperative complications. Using the Clavien–Dindo classification, they found that 62.7% of complications were minor (Grades I and II), 25.4% were major complications, and 11.9% resulted in death [33]. The rate of major complications and mortality was higher in their study. This difference may be explained by the longer follow-up period, as their study included complications occurring up to 30 days postoperatively, which could account for the observed variation.

### Strengths and limitations of the study

This study's strength lies in its prospective design, which allowed for the comprehensive recording of all complications that occurred during recovery. In contrast, a retrospective study design may overlook critical events or make them difficult to recall. However, a limitation of this study is that it assessed various types of surgical procedures and significant critical event types while encompassing a wide range of age groups. Therefore, it is recommended that further research focus on specific types of surgery. In addition, independent analyses of patients with Clavien–Dindo Grade III, IV, and V complications were not feasible due to the small sample size in these categories. This study was limited to evaluating immediate postoperative complications occurring in the PACU and did not consider delayed yet clinically significant events, such as pulmonary thromboembolism (PTE). Additionally, as this study was conducted at a single institution, conducting research in a multicenter setting with a larger sample size would be beneficial.

### Conclusion

Approximately 38% of patients experienced ≥1 critical event, mainly cardiovascular and respiratory complications. Patients who experienced intraoperative complications, had an ASA physical status of II to IV, under/overweight, and those who underwent prolonged anesthesia were at a relatively higher risk of developing critical events. Cardiovascular and respiratory-related critical events were the most frequently observed. Patients who experienced critical events also had longer stays in the post-anesthesia care unit (PACU).

### Recommendations

As confirmed in this study, patients who were attended by nurses with PACU training had a lower rate of critical events compared with those cared for by non-trained personnel. Therefore, we recommend that all PACU caregivers receive appropriate training in postoperative patient management. Hospitals should also establish standardized handover guidelines and implement structured handover checklists, as these measures can improve the quality of handovers and help minimize complications. Given that the majority of critical events occurred within the first hour after arrival in the PACU, all patients should be monitored closely and managed with heightened vigilance during this period. To obtain a national perspective on PACU-related complications and associated risk factors, we further recommend conducting multicenter studies.

### Supporting information

**S1 File. Data set.**
(XLSX)

# Acknowledgments

First and foremost, we would like to acknowledge Debre Markos University for providing us with the opportunity to conduct this research. We are grateful to the staff of the Department of Anesthesia at Debre Markos University for their invaluable support and insightful feedback throughout the study. Finally, we would like to thank the authors and researchers of the articles and online resources that we cited for their valuable contributions to this work.

# Author contributions

**Conceptualization:** Abebaw Misganaw, Enyew Fenta Mengistu.

**Data curation:** Abebaw Misganaw, Samrawit Nega Shiferaw.

**Formal analysis:** Abebaw Misganaw, Alaye Debas Ayenew, Baye Ashenef, Samrawit Nega Shiferaw.

**Funding acquisition:** Abebaw Misganaw, Enyew Fenta Mengistu, Samrawit Nega Shiferaw.

**Investigation:** Abebaw Misganaw, Alaye Debas Ayenew, Samrawit Nega Shiferaw, Getamesay Demelash Simegn.

**Methodology:** Abebaw Misganaw, Alaye Debas Ayenew, Netsanet Temesgen Ayenew, Enyew Fenta Mengistu, Baye Ashenef, Samrawit Nega Shiferaw, Getamesay Demelash Simegn.

**Project administration:** Abebaw Misganaw, Enyew Fenta Mengistu, Samrawit Nega Shiferaw.

**Resources:** Abebaw Misganaw, Baye Ashenef, Getamesay Demelash Simegn.

**Software:** Abebaw Misganaw, Alaye Debas Ayenew, Netsanet Temesgen Ayenew, Enyew Fenta Mengistu.

**Supervision:** Abebaw Misganaw, Netsanet Temesgen Ayenew, Baye Ashenef, Getamesay Demelash Simegn.

**Validation:** Abebaw Misganaw, Netsanet Temesgen Ayenew.

**Visualization:** Abebaw Misganaw, Netsanet Temesgen Ayenew, Baye Ashenef, Getamesay Demelash Simegn.

**Writing – original draft:** Abebaw Misganaw.

**Writing – review & editing:** Abebaw Misganaw, Netsanet Temesgen Ayenew.

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
