## [Decision Letter · Decision Letter 0]

4 Aug 2025

Dear Dr. Misganaw,

Thank you for submitting your manuscript to PLOS ONE. After careful consideration, we feel that it has merit but does not fully meet PLOS ONE’s publication criteria as it currently stands. Therefore, we invite you to submit a revised version of the manuscript that addresses the points raised during the review process.

We look forward to receiving your revised manuscript.

Kind regards,

Dereje Zewdu Assefa, BSc, MSc

Academic Editor

PLOS ONE

Journal Requirements:

2. Please amend the manuscript submission data (via Edit Submission) to include author Eniyew Fenta.

3. Please amend your authorship list in your manuscript file to include author Enyew Fenta Mengistu.

4. Please remove your figures from within your manuscript file, leaving only the individual TIFF/EPS image files, uploaded separately. These will be automatically included in the reviewers’ PDF.

5. We are unable to open your Supporting Information file [PACU complication.sav final.sav]. Please kindly revise as necessary and re-upload.

6. Please include captions for your Supporting Information files at the end of your manuscript, and update any in-text citations to match accordingly. Please see our Supporting Information guidelines for more information: http://journals.plos.org/plosone/s/supporting-information .

Additional Editor Comments:

The abstract section does not need to include a separate "Objective" heading, as the objective is already part of the background. Please revise and proofread the entire manuscript carefully, as there are numerous grammatical errors and typographical issues. Ensure consistency in terminology, especially regarding the use of "overweight" and "obesity," and avoid casual or interchangeable language.

Instead of stating, “Today, surgical procedures are accepted as one of the most important therapeutic methods in all societies worldwide,” consider rephrasing to: “Nowadays, surgical procedures are widely recognized as essential therapeutic interventions across all societies.”

Avoid redundancy and unnecessary detail. For example, the following sentence should be revised for clarity and brevity:

“A post-anesthesia care unit, often abbreviated as PACU and sometimes referred to as post-anesthesia recovery (PAR), is a vital part of hospitals. It is an area, normally attached to operating room suites, designed to provide care for patients recovering from general anesthesia, regional anesthesia, or local anesthesia.”

This should be simplified and written more precisely.

Remove the separate headings for “General” and “Specific Objectives” and instead integrate the study aim into the final paragraph of the introduction. Please refer to the PLOS ONE journal format and review previously published articles to guide your revisions.

The “Methods and Materials” section requires substantial language editing. For example, the sentence:

“Patients who were transferred directly from the operation room to an intensive care unit or ward excluded in the study,”

should be rephrased and justified, as major surgical procedures are often followed by direct transfer to ICU or ward, which may influence the incidence of critical events.

Although the authors state that a 50% prevalence was used due to the lack of similar studies in Ethiopia, there is in fact a related study with comparable findings. Please revise the justification accordingly.

Explain why seven data collectors and six supervisors were involved, as this may introduce variability and bias. Revise the data collection section to improve clarity and eliminate ambiguity.

Clarify why the pre-test was conducted at the same institution where the study took place.

The terminology regarding outcomes is inconsistent. Terms such as “PACU complications,” “critical events,” and “critical incidents” are used interchangeably. Please choose a single term and use it consistently throughout the manuscript.

In the study population, clarify the age range included. If the study spans a wide range (e.g., 3 to 86 years), consider whether this is clinically appropriate, given that complication rates and types likely differ across age groups.

Clarify whether your source population includes all surgical patients or only those undergoing major surgery. Also, assess whether including procedures performed under sedation in both major and minor ORs introduces clinical heterogeneity.

Table 2 needs revision, as some values (e.g., total sum of critical events by ASA class) exceed 100%. Verify and correct the data presentation.

Patients undergoing general, regional, or sedative anesthesia represent clinically distinct groups. Consider whether it is appropriate to analyze them together when measuring the incidence of postoperative critical events.

Further comments on the results and discussion sections will follow once the above major issues have been addressed.

Reviewers' comments:

Reviewer's Responses to Questions

**Comments to the Author**

1. Is the manuscript technically sound, and do the data support the conclusions?

Reviewer #1: Partly

Reviewer #2: Yes

Reviewer #3: Yes

2. Has the statistical analysis been performed appropriately and rigorously?

Reviewer #1: No

Reviewer #2: Yes

Reviewer #3: No

3. Have the authors made all data underlying the findings in their manuscript fully available?

Reviewer #1: Yes

Reviewer #2: Yes

Reviewer #3: Yes

4. Is the manuscript presented in an intelligible fashion and written in standard English?

Reviewer #1: No

Reviewer #2: Yes

Reviewer #3: Yes

Reviewer #1: Abstract; methodology (rephrase)

Background; paragraph 2:On the other hand, delayed effects of different anesthetic agents and muscle relaxants affects negatively the natural ability of the body to maintain physiological balance, which finally causes post-anesthetic critical incidents (critical events). (cite/ find reference)

last paragraph: Therefore the finding of this study will contribute in minimizing the doubt regarding the type of factors and common post-operative critical incidents. (rephrase)

Methodology; data collection; (rephrase)

Results: repeated several times in the text

Discussion: More of literature review. Discuss why did you get the results. Highlight the limitations and explain why eg: 2 hour delay, Improper passing over etc.

Reviewer #2: Well done. I love your research & how it was done. However, there are a couple issues I have with your work.

There are a lot of grammatical errors and poorly constructed sentences, especially in the abstract and background sections. Also, your work did not include If day cases or people already on admission had an effect on critical events in PACU.

Besides, I'd have loved it personally if the chart at the end of the work is brought into the result section. I'll have also loved if discharge scoring systems in PACU were examined in relation to critical incidence.

Reviewer #3: Reviewer Report for Manuscript PONE-D-25-27675

Summary :

This prospective study on PACU complications in a resource-limited Ethiopian hospital addresses a significant knowledge gap. While the topic is important and the design has strengths, major methodological, statistical, and reporting flaws preclude acceptance. Extensive revisions and re-analysis are essential.

Major Concerns :

1. Methodological Rigor :

- Sample size calculation using 50% prevalence lacks justification. Recalculate using literature-based estimates (e.g., 20-35%) and perform post-hoc power analysis.

- Critical event definitions are inconsistent and unvalidated (e.g., SpO₂<90% for >3 minutes; Clavien-Dindo applied to PACU without validation). Standardize terminology ("critical events" vs. "complications") and align with WHO/ANZCA guidelines. Report inter-rater reliability (κ-statistic).

- Data collection gaps : No standardized discharge criteria (e.g., Aldrete score), and missing data strategy unreported. Address "na" entries in complication timing/type (13% of events).

2. Statistical Validity :

- Regression models are inadequately developed :

- Age retained despite non-significance (p=0.31); no justification for variable inclusion.

- Multicollinearity unchecked (e.g., ASA-BMI-duration correlation likely).

- Full bivariable results omitted; sensitivity analyses missing.

- Incorrect incidence reporting :

- Cardiovascular events: 42% of *events* reported without *patient-level* incidence (21.3% of patients).

- Table 2 mislabels subgroup *incidence* (e.g., 69.5% in ≥65yrs) as "proportions," lacking 95% CIs.

- Data quality issues :

- Pediatric patients categorized with adult BMI.

- Inconsistent coding ("femal," "secere," variable "na" rates).

3. Reporting & Interpretation :

- Results :

- No CONSORT diagram for patient inclusion.

- Table 5: Wide CI for ASA4 (AOR=11.75, 95%CI:2.76-50.03) suggests instability; unaddressed.

- Discussion :

- Poor contextualization of high complication rates (37.9%) within resource limitations (e.g., 54.4% complications without nurse training).

- War-related caseload reduction ignored in generalizability.

- Limitations : Single-center design, observer bias, and selection bias (ICU-excluded patients) inadequately discussed.

Essential Revisions :

1. Statistical Re-analysis :

- Report patient-level incidence with 95% CIs for *all* complications (e.g., "21.3% [17.5–25.7%] of patients had cardiovascular events").

- Re-run regressions: Provide full bivariable results, check multicollinearity (VIF), validate models (ROC-AUC), and conduct sensitivity analyses.

- Post-hoc power analysis for key outcomes.

2. Methodological Corrections :

- Correct pediatric BMI categorization.

- Define missing data handling (e.g., "Multiple imputation used for 13.1% missing complication timings").

- Justify event definitions and validate Clavien-Dindo for PACU.

3. Reporting Compliance :

- Add 95% CIs to *every* incidence rate (Abstract, Results, Tables 2–5).

- Relabel Table 2: " Incidence of critical events, % (95% CI) " with footnote: "*% of subgroup patients with ≥1 event*."

- Ethical committee pproval number and data repository/access details (PLOS ONE requirement).

4. Contextual Enhancement :

- Discuss war impact on caseload/complications.

- Contrast findings with LMIC studies.

- Propose actionable interventions (e.g., nurse training program budget impact).

Long-term Recommendations :

- Multi-center validation.

- Standardized PACU discharge protocol development.

- Risk prediction model refinement.

Conclusion :

While this study highlights critical PACU safety challenges in resource-limited settings, methodological flaws and statistical misreporting undermine its contribution. I recommend partnering with a biostatistician to address these concerns. *Major revision is required before reconsideration*.

**Do you want your identity to be public for this peer review?** For information about this choice, including consent withdrawal, please see our Privacy Policy

Reviewer #1: **Yes:** Huda Zainal Abidin

Reviewer #2: **Yes:** Adefusi Temiloluwa Oluwakorede

Reviewer #3: **Yes:** Ali Afkhaminia

---

## [Author Response · Author response to Decision Letter 1]

3 Sep 2025

First of all, we would like to acknowledge the reviewers and editorial team for giving your time to review this manuscript. Then we have tried to reflect our responses as follow for reviewers and editors according to your concerns.

I. Editors

1. Please ensure that your manuscript meets PLOS ONE's style requirements, including those for file naming. The PLOS ONE style templates can be found at. Response: we have tried to be in line with PLOS ONE style.

2. Please amend the manuscript submission data (via Edit Submission) to include author Eniyew Fenta.

3. Please amend your authorship list in your manuscript file to include author Enyew Fenta Mengistu. Response: It’s been corrected.

4. Please remove your figures from within your manuscript file, leaving only the individual TIFF/EPS image files, uploaded separately. These will be automatically included in the reviewers’ PDF. Response: corrected.

5. We are unable to open your Supporting Information file [PACU complication.sav final.sav]. Please kindly revise as necessary and re-upload. Response: we re-uploaded the final version we used for analysis.

If the reviewer comments include a recommendation to cite specific previously published works, please review and evaluate these publications to determine whether they are relevant and should be cited. There is no requirement to cite these works unless the editor has indicated otherwise. Response: appropriate measurements were taken based on reviewers concern.

Additional Editor Comments:

The abstract section does not need to include a separate "Objective" heading, as the objective is already part of the background. Please revise and proofread the entire manuscript carefully, as there are numerous grammatical errors and typographical issues. Ensure consistency in terminology, especially regarding the use of "overweight" and "obesity," and avoid casual or interchangeable language. Response: as much as possible we tried to correct the issues you mentioned above.

Instead of stating, “Today, surgical procedures are accepted as one of the most important therapeutic methods in all societies worldwide,” consider rephrasing to: “Nowadays, surgical procedures are widely recognized as essential therapeutic interventions across all societies.” Response: it’s corrected.

Avoid redundancy and unnecessary detail. For example, the following sentence should be revised for clarity and brevity:

“A post-anesthesia care unit, often abbreviated as PACU and sometimes referred to as post-anesthesia recovery (PAR), is a vital part of hospitals. It is an area, normally attached to operating room suites, designed to provide care for patients recovering from general anesthesia, regional anesthesia, or local anesthesia.” Response: it’s corrected

This should be simplified and written more precisely.

Remove the separate headings for “General” and “Specific Objectives” and instead integrate the study aim into the final paragraph of the introduction. Please refer to the PLOS ONE journal format and review previously published articles to guide your revisions.

The “Methods and Materials” section requires substantial language editing. For example, the sentence:

“Patients who were transferred directly from the operation room to an intensive care unit or ward excluded in the study,” should be rephrased and justified, as major surgical procedures are often followed by direct transfer to ICU or ward, which may influence the incidence of critical events. Response: our focus is just to investigate the immediate post-operative critical events which means in the PACU. Otherwise if we included those patients transferred to ICU and ward, we belied that the incidence and associated factors could be changed from the PACU. Because the care given as well as the ratio of health care provider to patients in the ICU and ward if different from PACU, this affects our study. That is why we left ICU and ward patients.

Although the authors state that a 50% prevalence was used due to the lack of similar studies in Ethiopia, there is in fact a related study with comparable findings. Please revise the justification accordingly. Response: yes there are some studies in Ethiopia but the setup of these studies much better that our study area. Based on our previous observations, we perceived the PACU standard of our study hospital was poor. We discussed that to use 50% prevalence to investigate the situation with better sample size.

Explain why seven data collectors and six supervisors were involved, as this may introduce variability and bias. Revise the data collection section to improve clarity and eliminate ambiguity. Response: why we increased number of data collectors and supervisors, it is due to the nature of the study since we included all cases means emergencies and electives we have to collect all cases at nights, working days, and holidays. To do this we need relatively large number of data collectors and supervisors. It was difficult to small numbers of data collectors and supervisors. That is why we include relatively large numbers.

Clarify why the pre-test was conducted at the same institution where the study took place. Response: we consult experts and what they told us, you can do pretest in the same area but don’t included in the analysis. During our study period, there was active war thus both the internet and roads were locked. Therefore, we couldn’t do pretest in other institutions.

The terminology regarding outcomes is inconsistent. Terms such as “PACU complications,” “critical events,” and “critical incidents” are used interchangeably. Please choose a single term and use it consistently throughout the manuscript. Response: corrected

In the study population, clarify the age range included. If the study spans a wide range (e.g., 3 to 86 years), consider whether this is clinically appropriate, given that complication rates and types likely differ across age groups. Response: what we focused is just what the status of critical events in the PACU is. We want to investigate any types of critical events irrespective of age, sex, types of surgeries and the like. But we analyzed whether these factors have association or not with critical events. We considered age as independent factor and found that age was not associated with critical events.

Clarify whether your source population includes all surgical patients or only those undergoing major surgery. Also, assess whether including procedures performed under sedation in both major and minor ORs introduces clinical heterogeneity. Response: rephrased; the source population is all surgical patients who revived any types of anesthesia in the Major operation room.

Table 2 needs revision, as some values (e.g., total sum of critical events by ASA class) exceed 100%. Verify and correct the data presentation. Response: we have checked and may be the table is ambiguous but the proportion of critical events in subgroups (%) is the ratio of patients with critical events to the total number of patients in specific groups (eg in age below 15 years 18/53*100 =34%). This values can’t give 100%. These are just ratios in specific groups.

Patients undergoing general, regional, or sedative anesthesia represent clinically distinct groups. Consider whether it is appropriate to analyze them together when measuring the incidence of postoperative critical events. Response: we don’t believe that it is feasible to study the incidence and types of critical events by dividing by different types of anesthesia. It is better to check whether these types of anesthesia have association or not. What we do is just analyzed whether these types of anesthesia have association with critical events or not.

Further comments on the results and discussion sections will follow once the above major issues have been addressed.

II: Reviewers

Reviewer #1: Abstract; methodology (rephrase); Response: The whole paper including abstract has been re-written

Background; paragraph 2: On the other hand, delayed effects of different anesthetic agents and muscle relaxants affects negatively the natural ability of the body to maintain physiological balance, which finally causes post-anesthetic critical incidents (critical events). (cite/ find reference): Response; we have cited

last paragraph: Therefore the finding of this study will contribute in minimizing the doubt regarding the type of factors and common post-operative critical incidents. (rephrase); Response: Based on your suggestions we review grammatical and clarity issues carefully and it is rephrased

Methodology; data collection; (rephrase); Response: it is rephrased

Results: repeated several times in the text; Response: we tried to avoid unnecessary redundant words in the whole text.

Discussion: More of literature review. Discuss why you got the results. Highlight the limitations and explain why eg: 2 hour delay, Improper passing over etc. Response: we tried to address all issues.

Reviewer #2: Well done. I love your research & how it was done. However, there are a couple issues I have with your work: Response: Thank you very much dear reviewer for your appreciation.

There are a lot of grammatical errors and poorly constructed sentences, especially in the abstract and background sections. Also, your work did not include If day cases or people already on admission had an effect on critical events in PACU. Response: based on your suggestion we have reviewed carefully the issues of grammar and bad sentence constructions and we have taken measurements to make the study readable for scholars. Based on the protocol of the study hospital all patients who received anesthesia should be admitted in the recovery (PACU) except those patients directly sent to ICU. Although we have written ‘those patients directly transferred from operation room to ward and ICU as exclusion, but in practice there was no patient directly transferred to ward rather there were to ICU. Therefore, based on the study institution protocol all cases should be admitted to PACU before they discharged to ward or home. So, no cases missed once they exposed for anesthesia. We removed the word ‘ward’ in the manuscript since there were no patient transferred in this sentence ‘Patients who were transferred directly from the operation room to ward excluded in the study’.

Besides, I'd have loved it personally if the chart at the end of the work is brought into the result section. I'll have also loved if discharge scoring systems in PACU were examined in relation to critical incidence. Response: Thank you dear reviewer for your suggestions. Although we didn’t assess the scoring system of discharge but we haves assessed whether critical incidents prolong the timing of discharge and the result indicated that it significantly prolongs the PACU stay time. Scholars reported that prolonged stay in the PACU has economic burdens both for patents and hospitals.

Reviewer #3: Reviewer Report for Manuscript PONE-D-25-27675

Summary :

This prospective study on PACU complications in a resource-limited Ethiopian hospital addresses a significant knowledge gap. While the topic is important and the design has strengths, major methodological, statistical, and reporting flaws preclude acceptance. Extensive revisions and re-analysis are essential.

Major Concerns :

1. Methodological Rigor :

- Sample size calculation using 50% prevalence lacks justification. Recalculate using literature-based estimates (e.g., 20-35%) and perform post-hoc power analysis. Response: why we didn’t use previous studies is, we believed the setup of our study area far different than the others. Based on our previous observations, the study area lacks ether local or international guidelines like WHO, ASA… etc. therefore, our expectation was the incidence rate greater than we found in this study. So, why we used 50% prevalence is just to investigate critical events with larger sample size.

- Critical event definitions are inconsistent and unvalidated (e.g., SpO₂<90% for >3 minutes; Clavien-Dindo applied to PACU without validation). Standardize terminology ("critical events" vs. "complications") and align with WHO/ANZCA guidelines. Report inter-rater reliability (κ-statistic). Response: we used critical events from different studies as we cited and we didn’t used Clavien-Dindo’s report for data collection rather we just used for comparison once we analyzed. It’s to know in which grade of Clavien-Dindo grading our finding was lied. Regarding terminology, ANZCA and many literatures used both terms but for preventable serious sudden complications used either critical incidents or events. Therefore, in the PACU the types of complications happened are most of the time are serious unless managed early. We made consistent the terms and used the term critical events.

- Data collection gaps : No standardized discharge criteria (e.g., Aldrete score), and missing data strategy unreported. Address "na" entries in complication timing/type (13% of events). Response: discharge criteria was not in our study objectives. Our objective was to study critical events. But as a secondary objective we assessed whether there was association between critical events and fulfillment of patients to discharge criteria at 2 hours using Aldrete score. As a study we used Aldrete score but the study institution hadn’t it. Regarding the missing data management, although it is possible to use different strategy of missing data management like using mean, median, or mode but we authors agreed to reject the data which contains missing content and replace with other data until the calculated data fulfilled to improve the accuracy of data. Luckily we had no any missing content in our data.

2. Statistical Validity :

- Regression models are inadequately developed: Response: we have discussed with biostatisticians after your suggestions and they gave us binary logistic regression is appropriate model to analyze our data. We checked all the assumptions of logistic regression and it fitted.

- Age retained despite non-significance (p=0.31); no justification for variable inclusion. : Response: sorry for making the table ambiguous. But the COR p-value (bivariable analysis) of age was below 0.2 and the p-value set in the table 5 is only for AOR

- Multicollinearity unchecked (e.g., ASA-BMI-duration correlation likely). Response: we have checked the assumption of logistic regression and the data fulfils the assumption with VIF value of below 2.3, tolerance value greater than 0.4, and inter-correlation of spearman correlation value below 0.7. Although logically BMI affects ASA status in our study the statistics indicate that there is no correlation between variables.

- Full bivariable results omitted; sensitivity analyses missing. Response: we left bivarible result from the final table to avoid the condensed table. But after your suggestion we observed as it is ambiguous and we incorporated the Crude odd ratio in table 5. Table 5 contains those variables that fulfill COR p-value of < 0.2 and then run for multivariable logistic regression.

- Incorrect incidence reporting :

- Cardiovascular events: 42% of *events* reported without *patient-level* incidence (21.3% of patients). Response: corrected

- Table 2 mislabels subgroup *incidence* (e.g., 69.5% in ≥65yrs) as "proportions," lacking 95% CIs. Response: table 2 to table 4 are frequency tables and the labeling is corrected and the frequency of critical events in each sub groups were inserted.

- Data quality issues :

- Pediatric patients categorized with adult BMI. Response: we used the BMI on Age- and Sex-Specific Charts of CDC for children then we included the underweight in <18.5 group, the healthy weight in 18.5-24.9, and overweight & obese in >24.9 groups. But as

---

## [Decision Letter · Decision Letter 1]

13 Oct 2025

Dear Dr. Misganaw,

Thank you for submitting your manuscript to PLOS ONE. After careful consideration, we feel that it has merit but does not fully meet PLOS ONE’s publication criteria as it currently stands. Therefore, we invite you to submit a revised version of the manuscript that addresses the points raised during the review process.

We look forward to receiving your revised manuscript.

Kind regards,

Dereje Zewdu Assefa, BSc, MSc

Academic Editor

PLOS ONE

Journal Requirements:

**Additional Editor Comments:**

In the abstract section

In methods of abstract section please revise it in this way “An institution-based prospective cross-sectional study was conducted from June 1, 2024, to September 30, 2024.” Revise as follows “Analysis was conducted using bivariable and multivariable logistic regression as needed.”

In the results section, you stated that "Out of 422 operations, 37.9% of patients developed about 214 complications." Please consider rephrasing this statement for clarity.

In the conclusion, revise as follows: “Patients who had intraoperative complications, higher ASA status, unhealthy body weight, and those who received anesthesia for a prolonged duration were relatively at higher risk of developing critical events.” Please replace the term "unhealthy body weight" with the appropriate term using BMI.

Change the heading title from "Background" to "Introduction" at the end of the abstract; it should begin as "Introduction," not "Background."

In the introduction section, please include references that support the evidence you are using as much as possible. I've noticed that some statements have only one or two citations, despite being quite broad.

The language and grammar have been significantly improved; however, it still needs revision. Please read it carefully line by line and use appropriate transition terms to connect each statement throughout the entire document.

Revise as stated “In developing countries, including Ethiopia, most PACU setups do not meet WHO standards, and some hospitals lack a PACU altogether, sending patients directly from the operating room to the ward.”

You mentioned that "Although several studies have been published on complications in the PACU, there is limited information in Africa." However, you need to provide a reference for this statement.

Please refrain from providing detailed lists of independent and dependent variables. Instead, use statements to describe them.

Please provide details on how the data were transitioned from bivariate to multivariate analysis, as well as how multicollinearity was assessed in the data analysis section.

Please provide details in the limitations section. For example, various types of surgical procedures and significant critical event types were assessed, and a wide range of age groups was included. It is recommended that specific research focusing on particular types of surgery be conducted.

Reviewers' comments:

Reviewer's Responses to Questions

**Comments to the Author**

Reviewer #2: All comments have been addressed

Reviewer #3: All comments have been addressed

2. Is the manuscript technically sound, and do the data support the conclusions?

Reviewer #2: Yes

Reviewer #3: Yes

3. Has the statistical analysis been performed appropriately and rigorously?

Reviewer #2: Yes

Reviewer #3: Yes

4. Have the authors made all data underlying the findings in their manuscript fully available?

Reviewer #2: Yes

Reviewer #3: Yes

5. Is the manuscript presented in an intelligible fashion and written in standard English?

Reviewer #2: Yes

Reviewer #3: Yes

Reviewer #2: This research topic is very interesting and important also. I'm also happy with the improvement in the review of the article as compared to the initial draft, especially as regards grammar and construct. I appreciate the correlations between critical incidents and ASA classification, trained personnel, BMI and other criteria. I also appreciate the "Recommendation" section because the article did outline the need for a proper PACU handover criteria and training of PACU personnel to help with prevention and management of complications.

However, I'll like to query the choice of words and terminologies such as Tachyarrythmia and Bradyarrythmia over tachycardia and bradycardia. Hypertension with Systolic Blood Pressure >180mmHg is according to what criteria?

When patients were finally discharged from PACU, what discharge criteria was used?

Besides, your study did not account for any difference in critical incidence between Day cases and in patients, if there are any.

Lastly, a few grammar and construct adjustments could be made to perfect the article. Nonetheless, it's a work well done!

Reviewer #3: General Comments

This is an important prospective study addressing perioperative safety in a resource-limited setting. The topic is highly relevant, particularly given the paucity of data on PACU outcomes in sub-Saharan Africa. The revised version has improved substantially; however, there remain issues in clarity, structure, reporting, and methodological detail that should be addressed before acceptance. Below, I provide detailed comments for the authors and editorial team.

Major Concerns

Abstract

The sentence “A P-value <0.05 was considered statistically significant” should be removed from the abstract; such technical details belong in the Methods.

The conclusion is vague: “A significant number of patients suffered…”. Please quantify (“Approximately 38% of patients experienced ≥1 critical event, mainly cardiovascular and respiratory complications.”).

Background

The abbreviation ASA is used without definition. Please spell out in full at first mention (American Society of Anesthesiologists physical status classification), then abbreviate.

Methods

Independent/Dependent Variables: These are listed as bullet points (p. 5). For PLOS ONE style, rephrase into sentences/paragraphs (e.g., “Independent variables included age, sex, ASA status, BMI, type of anesthesia…”).

Statistical Model Fit: The manuscript mentions Omnibus and Hosmer–Lemeshow tests but does not explain them. Please describe their purpose and null hypotheses (Omnibus: H₀ = no predictors improve the model; Hosmer–Lemeshow: H₀ = model fits observed data). Interpret the reported p-values (e.g., p=0.085 indicates no evidence of poor fit).

Adjustment Strategy: Clarify that only variables with p<0.20 in bivariable analysis were retained for multivariable regression, and state explicitly that other covariates were excluded.

Ethics: The ethics section mentions approval but no code/number. Please provide the approval code, or explicitly state that the committee did not issue one.

Clavien–Dindo Grades: Consider a separate regression analysis restricted to Grade III–V complications to identify predictors of severe, life-threatening events.

Results – Tables

Table 2–4: The column header “Proportion of critical events in subgroups” should be replaced with “Incidence of critical events (% with ≥1 event, 95% CI)”. Confidence intervals should be added.

Narrative Clarity: Avoid vague terms like “nearly equal” (p. 11); report actual numbers/percentages.

Event Reporting: For each complication (hypotension, shock, arrhythmia, etc.), present both counts and percentages (e.g., “41 cases of hypotension, 19.2%”).

Nurse Training Comparison: Where differences are noted (trained vs. untrained), present percentages and statistical results (p-values/AOR) in the text, not only in tables.

Table 5:

Add a column for p-values corresponding to CORs.

Clarify which covariates were included in AOR adjustment.

Figures – Flow Diagram

The boxes require alignment correction for readability.

Excluded patients are not quantified. Please add the total n of excluded patients with reasons.

The n = 422 included label should be placed adjacent to “Patients included in the study,” not before.

The final node should branch into two groups: “Patients with ≥1 critical event (n = …)” and “Patients without critical events (n = …)”. This would better communicate study outcomes.

Limitations

The limitations section should note that the study only assessed immediate PACU complications and could not capture delayed but clinically important events such as pulmonary thromboembolism (PTE). Acknowledge that this is a serious limitation of the PACU-only timeframe.

Minor Concerns

Replace the phrase “post-anesthesia complications also known as critical incidents” with more consistent terminology (“critical events”) throughout.

Ensure that percentages are consistently reported alongside raw numbers across Results and Discussion.

Revise wording such as “nearly equal” or “not surprising” with precise, scientific expressions.

Flow and grammar should be further polished (e.g., “a P-value <0.05 was consider” → “considered”).

Conclusion

This manuscript addresses an under-researched but highly important area of perioperative safety in low-resource settings. However, improvements are needed in reporting clarity, statistical transparency, and figure/table presentation. Addressing the above concerns—particularly the Abstract, Results tables, flow diagram, and model explanation—will substantially strengthen the manuscript and make it more suitable for publication.

**Do you want your identity to be public for this peer review?** For information about this choice, including consent withdrawal, please see our Privacy Policy

Reviewer #2: No

Reviewer #3: **Yes:** Ali Afkhaminia

---

## [Author Response · Author response to Decision Letter 2]

14 Nov 2025

First, we would like to thank the reviewers and the editorial team for taking the time to review our manuscript. We have addressed your concerns as follows in our responses to the reviewers and editors.

Additional Editor Comments:

In the abstract section

In methods of abstract section please revise it in this way “An institution-based prospective cross-sectional study was conducted from June 1, 2024, to September 30, 2024.” Revise as follows “Analysis was conducted using bivariable and multivariable logistic regression as needed.” Response: corrected

In the results section, you stated that "Out of 422 operations, 37.9% of patients developed about 214 complications." Please consider rephrasing this statement for clarity.

In the conclusion, revise as follows: “Patients who had intraoperative complications, higher ASA status, unhealthy body weight, and those who received anesthesia for a prolonged duration were relatively at higher risk of developing critical events.” Please replace the term "unhealthy body weight" with the appropriate term using BMI. Response: corrected

Change the heading title from "Background" to "Introduction" at the end of the abstract; it should begin as "Introduction," not "Background." Response: corrected

In the introduction section, please include references that support the evidence you are using as much as possible. I've noticed that some statements have only one or two citations, despite being quite broad. Response: we considered your suggestions

The language and grammar have been significantly improved; however, it still needs revision. Please read it carefully line by line and use appropriate transition terms to connect each statement throughout the entire document. Response: we considered your suggestions

Revise as stated “In developing countries, including Ethiopia, most PACU setups do not meet WHO standards, and some hospitals lack a PACU altogether, sending patients directly from the operating room to the ward.” Response: corrected

You mentioned that "Although several studies have been published on complications in the PACU, there is limited information in Africa." However, you need to provide a reference for this statement. Response: we have cited based on your recommendation

Please refrain from providing detailed lists of independent and dependent variables. Instead, use statements to describe them. Response: corrected

Please provide details on how the data were transitioned from bivariate to multivariate analysis, as well as how multicollinearity was assessed in the data analysis section. Response: corrected

Please provide details in the limitations section. For example, various types of surgical procedures and significant critical event types were assessed, and a wide range of age groups was included. It is recommended that specific research focusing on particular types of surgery be conducted. Response: corrected

Reviewer #2: This research topic is very interesting and important also. I'm also happy with the improvement in the review of the article as compared to the initial draft, especially as regards grammar and construct. I appreciate the correlations between critical incidents and ASA classification, trained personnel, BMI and other criteria. I also appreciate the "Recommendation" section because the article did outline the need for a proper PACU handover criteria and training of PACU personnel to help with prevention and management of complications.

However, I'll like to query the choice of words and terminologies such as Tachyarrythmia and Bradyarrythmia over tachycardia and bradycardia. Hypertension with Systolic Blood Pressure >180mmHg is according to what criteria? Response: we used the definition set in the anesthesia book of Morgan & Mikhail’s Clinical Anesthesiology, 7th ed. McGraw-Hill, 2022. And also we used previous studies cited in our paper. And we corrected the terminologies as your suggestions to tachycardia and bradycardia.

When patients were finally discharged from PACU, what discharge criteria was used? Response: we assessed whether there was association between critical events and fulfillment of patients to discharge criteria at 2 hours using Aldrete score. As a study we used Aldrete score but the study institution hadn’t it. Whether professionals planned to discharge patients from PACU or not we assessed all patients after 2 hours of waiting in PACU. But in practical most of patients discharged from PACU after more than 5 hours waiting due to the absence of discharge protocol in the institution.

Besides, your study did not account for any difference in critical incidence between Day cases and in patients, if there are any. Response: although our inclusion criteria allows to include day cases in our study, but there were no day-case procedures in our study since the hospital had no day-case schedule till end of our study period.

Lastly, a few grammar and construct adjustments could be made to perfect the article. Nonetheless, it's a work well done! Response: thank you for your appreciation dear reviewer for your constructive comments to make this article an interesting and we tried to correct the grammar and other concerns.

Reviewer #3: General Comments

This is an important prospective study addressing perioperative safety in a resource-limited setting. The topic is highly relevant, particularly given the paucity of data on PACU outcomes in sub-Saharan Africa. The revised version has improved substantially; however, there remain issues in clarity, structure, reporting, and methodological detail that should be addressed before acceptance. Below, I provide detailed comments for the authors and editorial team.

Major Concerns

Abstract

The sentence “A P-value <0.05 was considered statistically significant” should be removed from the abstract; such technical details belong in the Methods. Response: corrected

The conclusion is vague: “A significant number of patients suffered…”. Please quantify (“Approximately 38% of patients experienced ≥1 critical event, mainly cardiovascular and respiratory complications.”). Response: we have corrected based on your suggestions

Background

The abbreviation ASA is used without definition. Please spell out in full at first mention (American Society of Anesthesiologists physical status classification), then abbreviate. Response: we have corrected.

Methods

Independent/Dependent Variables: These are listed as bullet points (p. 5). For PLOS ONE style, rephrase into sentences/paragraphs (e.g., “Independent variables included age, sex, ASA status, BMI, type of anesthesia…”). Response: it’s corrected based on your suggestions.

Statistical Model Fit: The manuscript mentions Omnibus and Hosmer–Lemeshow tests but does not explain them. Please describe their purpose and null hypotheses (Omnibus: H₀ = no predictors improve the model; Hosmer–Lemeshow: H₀ = model fits observed data). Interpret the reported p-values (e.g., p=0.085 indicates no evidence of poor fit). Response: we have corrected based on your suggestions.

Adjustment Strategy: Clarify that only variables with p<0.20 in bivariable analysis were retained for multivariable regression, and state explicitly that other covariates were excluded. Response: we clarified in this way ‘Only variables with a p-value < 0.20 in the bivariable analysis were included in the multivariable regression model, while covariates with p ≥ 0.20 were excluded’.

Ethics: The ethics section mentions approval but no code/number. Please provide the approval code, or explicitly state that the committee did not issue one. Response: we have ethical approval letter with the code of R/C/S/D/357/01/18. And, if we are requested we can attach the letter. We corrected and mentioned in ethics part.

Clavien–Dindo Grades: Consider a separate regression analysis restricted to Grade III–V complications to identify predictors of severe, life-threatening events. Response: we have faced difficulties to analyses the association between dependent variables and covariates in the samples who have Clavien–Dindo class III and IV. It is not fulfilling the assumption (large covariates with small sample).

Results – Tables

Table 2–4: The column header “Proportion of critical events in subgroups” should be replaced with “Incidence of critical events (% with ≥1 event, 95% CI)”. Confidence intervals should be added. Response: corrected

Narrative Clarity: Avoid vague terms like “nearly equal” (p. 11); report actual numbers/percentages. Response: corrected.

Event Reporting: For each complication (hypotension, shock, arrhythmia, etc.), present both counts and percentages (e.g., “41 cases of hypotension, 19.2%”). Response: corrected

Nurse Training Comparison: Where differences are noted (trained vs. untrained), present percentages and statistical results (p-values/AOR) in the text, not only in tables. Response: we have mentioned it in text addition to the table in page 13.

Table 5:

Add a column for p-values corresponding to CORs. Response: corrected

Clarify which covariates were included in AOR adjustment. Response: we have mentioned variables which included in AOR adjustment in text part.

Figures – Flow Diagram

The boxes require alignment correction for readability.

Excluded patients are not quantified. Please add the total n of excluded patients with reasons.

The n = 422 included label should be placed adjacent to “Patients included in the study,” not before.

The final node should branch into two groups: “Patients with ≥1 critical event (n = …)” and “Patients without critical events (n = …)”. This would better communicate study outcomes.

Response: re-adjusted

Limitations

The limitations section should note that the study only assessed immediate PACU complications and could not capture delayed but clinically important events such as pulmonary thromboembolism (PTE). Acknowledge that this is a serious limitation of the PACU-only timeframe. Response: we accept all of your comments as it is. And we accept this very important comment also and we incorporated in the limitation section.

Minor Concerns

Replace the phrase “post-anesthesia complications also known as critical incidents” with more consistent terminology (“critical events”) throughout. Response: rephrased

Ensure that percentages are consistently reported alongside raw numbers across Results and Discussion. Response: we have checked it

Revise wording such as “nearly equal” or “not surprising” with precise, scientific expressions. Response: we have corrected.

Flow and grammar should be further polished (e.g., “a P-value <0.05 was consider” → “considered”). Response: corrected

---

## [Decision Letter · Decision Letter 2]

22 Dec 2025

Dear Dr. Misganaw,

Thank you for submitting your manuscript to PLOS ONE. After careful consideration, we feel that it has merit but does not fully meet PLOS ONE’s publication criteria as it currently stands. Therefore, we invite you to submit a revised version of the manuscript that addresses the points raised during the review process.

We look forward to receiving your revised manuscript.

Kind regards,

Dereje Zewdu Assefa, BSc, MSc

Academic Editor

PLOS One

Journal Requirements:

**Additional Editor Comments:**

The abstract should be short and precise. Please revise the background section of your abstract as follows: ‘Surgery and anaesthesia can disrupt normal physiological function through surgical stress and residual anaesthetic effects, increasing the risk of post-anaesthetic complications, known as critical incidents. This study aimed to determine the incidence of critical events in the post-anaesthesia care unit at Debre Markos Comprehensive Specialized Hospital, Ethiopia.”

On the results of the abstract section, please provide OR (odds ratio) of associated factors.

In the conclusion section, you have mentioned that ASA status was associated with the risk of developing critical events. Please be specific about which group of ASA status is associated with complications.

In the conclusion, it is better to mention under/overweight instead of describing them separately.

In the introduction section, you don’t need to provide a dearth of information; please be specific to your study and avoid repetition, and try to summarise long statements.

In the methods section, instead of using the study variable sub-heading, change it to the study outcomes and write it as follows:

The main outcome measure was critical events in the PACU, which was defined as… the point mentioned in the operational definitions, and delete the statement included in the operational definitions of critical outcomes. Then, include the secondary outcomes accordingly.

Rewrite your limitations and strengths section as follows: This study's strength lies in its prospective design, which allowed for the comprehensive recording of all complications that occurred during recovery. In contrast, a retrospective study design may overlook critical events or make them difficult to recall. However, a limitation of this study is that it assessed various types of surgical procedures and significant critical event types while encompassing a wide range of age groups. Therefore, it is recommended that further research focus on specific types of surgery. This study was limited to evaluating immediate postoperative complications occurring in the PACU and did not consider delayed yet clinically significant events, such as pulmonary thromboembolism (PTE). Additionally, as this study was conducted at a single institution, conducting research in a multicentre setting with a larger sample size would be beneficial.

Reviewers' comments:

Reviewer's Responses to Questions

**Comments to the Author**

Reviewer #3: All comments have been addressed

2. Is the manuscript technically sound, and do the data support the conclusions?

Reviewer #3: Yes

3. Has the statistical analysis been performed appropriately and rigorously?

Reviewer #3: Yes

4. Have the authors made all data underlying the findings in their manuscript fully available?

Reviewer #3: Yes

5. Is the manuscript presented in an intelligible fashion and written in standard English?

Reviewer #3: Yes

Reviewer #3: . Abstract:

Result: Clarify the first sentence for precision. Instead of "Out of 422 patients, 37.9% of patients developed about 214 complications," consider:

"Of the 422 patients, 160 (37.9%) experienced one or more critical events, with a total of 214 complications recorded."

Conclusion: Ensure consistency. The phrase "unhealthy body weight" in the conclusion should be explicitly tied to the BMI categories used (overweight/underweight).

2. Introduction:

Flow: Improve connectivity between sentences. Use transition words (e.g., Furthermore, Consequently, However) to guide the reader through the rationale.

Citations: Verify that the statement about limited data in Africa is directly supported by the newly added references (e.g., Blaise Pascal FN et al. 2021).

3. Methods:

Ethics: The approval code R/C/S/D/357/01/18 is now included, which is good.

Clarity: The explanation for not analyzing Clavien-Dindo Grade III-V separately (sample size limitations) is acceptable but should be stated clearly in the limitations section if not already.

4. Results:

Language: Replace informal phrases. For example:

"Even though it’s not surprising..." → "As anticipated, the incidence..."

"The proportions... were nearly equal..." → "The proportions were similar across groups, ranging from 35% to 38.4%."

Narrative: When discussing Table 5 in the text, explicitly state the key findings (e.g., "Patients attended by untrained nurses had over three times the odds of experiencing a critical event (AOR=3.15...).")

5. Discussion:

Tone: Maintain an objective, scientific tone. Avoid speculative phrasing like "We couldn’t provide the possible reasons..." Instead, state: "The reason for this discrepancy is not clear from our data but may relate to differences in case mix or definitions."

Structure: When comparing studies, first state the agreement or disagreement, then provide the comparative data, and finally offer a brief, plausible reason for the difference.

6. Language & Grammar (Final Polish):

Conduct a meticulous line-by-line edit focusing on:

Article Use: Ensure correct use of "a," "an," and "the."

Prepositions: Check "in," "on," "at," "for," etc.

Subject-Verb Agreement: (e.g., "The data were analyzed").

Plurals: (e.g., "complications").

Typos: "Sever hypoxemia" → "Severe hypoxemia"; "statistically significant" (not "statically").

7. Figures & Tables:

Flowchart (Figure 1): Ensure it matches the description in the "Response to Reviewers":

Quantify excluded patients (n=11).

Place "n = 422" label next to "Patients included in the study".

Final branch clearly shows: "Patients with ≥1 critical event (n=160)" and "Patients without critical events (n=262)".

Submission Checklist:

Before final submission, verify:

The Response to Reviewers letter is complete, polite, and addresses every point raised by each reviewer and the editor.

The Data Availability Statement in the submission system is accurate.

All supporting files (dataset, checklist, ethical approval) are uploaded.

The manuscript text is the "clean" version with all track changes accepted and no comment bubbles.

Conclusion

The manuscript is in its final stages. By implementing these focused revisions for clarity, precision, and language polish, you will significantly strengthen it for publication. The study provides valuable, actionable insights for improving PACU care in similar resource-limited settings.

**Do you want your identity to be public for this peer review?** For information about this choice, including consent withdrawal, please see our Privacy Policy

Reviewer #3: **Yes:** Ali Afkhaminia

---

## [Author Response · Author response to Decision Letter 3]

29 Dec 2025

Responses to editors' and reviewers' comment

First of all, we would like to acknowledge the reviewers and editorial team for giving your time to review this manuscript. Then we have tried to reflect our responses as follow for reviewers and editors according to your concerns.

I. Editors

Additional Editor Comments:

The abstract should be short and precise. Please revise the background section of your abstract as follows: ‘Surgery and anaesthesia can disrupt normal physiological function through surgical stress and residual anaesthetic effects, increasing the risk of post-anaesthetic complications, known as critical incidents. This study aimed to determine the incidence of critical events in the post-anaesthesia care unit at Debre Markos Comprehensive Specialized Hospital, Ethiopia.” Response: corrected as your suggestions

On the results of the abstract section, please provide OR (odds ratio) of associated factors. Resposne: corrected

In the conclusion section, you have mentioned that ASA status was associated with the risk of developing critical events. Please be specific about which group of ASA status is associated with complications. Response: corrected based on you suggestion

In the conclusion, it is better to mention under/overweight instead of describing them separately. Response: corrected

In the introduction section, you don’t need to provide a dearth of information; please be specific to your study and avoid repetition, and try to summarise long statements. Response: we have tried to avoid unnecessary repetitions. Thank you.

In the methods section, instead of using the study variable sub-heading, change it to the study outcomes and write it as follows:

The main outcome measure was critical events in the PACU, which was defined as… the point mentioned in the operational definitions, and delete the statement included in the operational definitions of critical outcomes. Then, include the secondary outcomes accordingly. Response: we have corrected it

Rewrite your limitations and strengths section as follows: This study's strength lies in its prospective design, which allowed for the comprehensive recording of all complications that occurred during recovery. In contrast, a retrospective study design may overlook critical events or make them difficult to recall. However, a limitation of this study is that it assessed various types of surgical procedures and significant critical event types while encompassing a wide range of age groups. Therefore, it is recommended that further research focus on specific types of surgery. This study was limited to evaluating immediate postoperative complications occurring in the PACU and did not consider delayed yet clinically significant events, such as pulmonary thromboembolism (PTE). Additionally, as this study was conducted at a single institution, conducting research in a multicentre setting with a larger sample size would be beneficial. Response: Thank you very much! For your great concerns to strengthen this manuscript for publication. We have done it.

II. Reviewers

We are grateful to you for your insightful comments and constructive feedback, which significantly enhance the clarity and rigor of the manuscript.

1. Result: Clarify the first sentence for precision. Instead of "Out of 422 patients, 37.9% of patients developed about 214 complications," consider:

"Of the 422 patients, 160 (37.9%) experienced one or more critical events, with a total of 214 complications recorded."

Response: we made a correction thank you

Conclusion: Ensure consistency. The phrase "unhealthy body weight" in the conclusion should be explicitly tied to the BMI categories used (overweight/underweight). Response: corrected

2. Introduction:

Flow: Improve connectivity between sentences. Use transition words (e.g., Furthermore, Consequently, However) to guide the reader through the rationale.

Citations: Verify that the statement about limited data in Africa is directly supported by the newly added references (e.g., Blaise Pascal FN et al. 2021). Response: we have rechecked and made it appropriate

3. Methods:

Ethics: The approval code R/C/S/D/357/01/18 is now included, which is good.

Clarity: The explanation for not analyzing Clavien-Dindo Grade III-V separately (sample size limitations) is acceptable but should be stated clearly in the limitations section if not already. Response: The number of patients with Clavien-Dindo Grade III-V complications is not controllable. But it might be increased and be adequate when originally the sample size is very large. And we mentioned it in a limitation section

4. Results:

Language: Replace informal phrases. For example:

"Even though it’s not surprising..." → "As anticipated, the incidence..." Response: we have corrected

"The proportions... were nearly equal..." → "The proportions were similar across groups, ranging from 35% to 38.4%." Response: we made corrections

Narrative: When discussing Table 5 in the text, explicitly state the key findings (e.g., "Patients attended by untrained nurses had over three times the odds of experiencing a critical event (AOR=3.15...).") Response: we have corrected based on your suggestion.

5. Discussion:

Tone: Maintain an objective, scientific tone. Avoid speculative phrasing like "We couldn’t provide the possible reasons..." Instead, state: "The reason for this discrepancy is not clear from our data but may relate to differences in case mix or definitions." Response: Thank you dear reviewer for giving us such interesting lesson. Every statement you wrote were meaningful for us. We have corrected

Structure: When comparing studies, first state the agreement or disagreement, then provide the comparative data, and finally offer a brief, plausible reason for the difference.

Response: we have tried our best to make based on your suggestions

6. Language & Grammar (Final Polish):

Conduct a meticulous line-by-line edit focusing on:

Article Use: Ensure correct use of "a," "an," and "the."

Prepositions: Check "in," "on," "at," "for," etc.

Subject-Verb Agreement: (e.g., "The data were analyzed").

Plurals: (e.g., "complications").

Typos: "Sever hypoxemia" → "Severe hypoxemia"; "statistically significant" (not "statically").

Response: We have checked the whole document to address the issues of articles, prepositions, grammar, plurals, and typos. We made corrections as much as we can.

7. Figures & Tables:

Flowchart (Figure 1): Ensure it matches the description in the "Response to Reviewers":

Quantify excluded patients (n=11).

Place "n = 422" label next to "Patients included in the study".

Final branch clearly shows: "Patients with ≥1 critical event (n=160)" and "Patients without critical events (n=262)".

Response: we have adjusted based on your suggestions

Submission Checklist: Response: We would like to express our sincere gratitude to you who provided us a detailed and constructive feedback through multiple rounds of review. Your valuable comments, professional insights, and shared research experience greatly enhanced the quality, clarity, and scientific rigor of this manuscript.

We have tried to make this manuscript based on this checklist

Before final submission, verify:

The Response to Reviewers letter is complete, polite, and addresses every point raised by each reviewer and the editor.

The Data Availability Statement in the submission system is accurate.

All supporting files (dataset, checklist, ethical approval) are uploaded.

The manuscript text is the "clean" version with all track changes accepted and no comment bubbles.

---

## [Editor Report · Decision Letter 3]

17 Feb 2026

Dear Dr. Misganaw,

Thank you for submitting your manuscript to PLOS ONE. After careful consideration, we feel that it has merit but does not fully meet PLOS ONE’s publication criteria as it currently stands. Therefore, we invite you to submit a revised version of the manuscript that addresses the points raised during the review process.

We look forward to receiving your revised manuscript.

Kind regards,

Dereje Zewdu Assefa, BSc, MSc

Academic Editor

PLOS One

Journal Requirements:

Additional Editor Comments:

Please change the title "Incidence of Critical Events in the Post-Anaesthesia Care Unit at a Resource-Limited Setting in Debre Markos, Northwest Ethiopia" to lower case. Additionally, please proofread for any typos once again.

---

## [Author Response · Author response to Decision Letter 4]

25 Feb 2026

There is no attachment of the reviewers' document. And we have addressed concerns raised by editors. Which are the manuscript title correcting typos, and making figures based on the PLOS ONE standard. All these are addressed.

---

## [Editor Report · Decision Letter 4]

11 Mar 2026

Incidence of Critical Events in the Post-Anesthesia Care Unit at a Resource-Limited Setting in Debre Markos, Northwest Ethiopia

PONE-D-25-27675R4

Dear Dr. Misganaw,

We’re pleased to inform you that your manuscript has been judged scientifically suitable for publication and will be formally accepted for publication once it meets all outstanding technical requirements.

Kind regards,

Dereje Zewdu Assefa, BSc, MSc

Academic Editor

PLOS One
---

## [Editor Report · Acceptance letter]

PONE-D-25-27675R4

PLOS One

Dear Dr. Misganaw,

I'm pleased to inform you that your manuscript has been deemed suitable for publication in PLOS One. Congratulations! Your manuscript is now being handed over to our production team.

Kind regards,

on behalf of

Professor Dereje Zewdu Assefa

Academic Editor

PLOS One